# Diagnosis and Treatment of Children with a Radiological Fat Pad Sign without Visible Elbow Fracture Vary Widely: An International Online Survey and Development of an Objective Definition

**DOI:** 10.3390/children9070950

**Published:** 2022-06-25

**Authors:** Maximiliaan A. Poppelaars, Denise Eygendaal, Bertram The, Iris van Oost, Christiaan J. A. van Bergen

**Affiliations:** 1Department of Orthopaedic Surgery, Amphia Hospital, 4818 CK Breda, The Netherlands; bthe@amphia.nl (B.T.); cvanbergen@amphia.nl (C.J.A.v.B.); 2Department of Orthopaedic and Sports Medicine, Erasmus Medical Centre, 3015 GD Rotterdam, The Netherlands; d.eygendaal@erasmusmc.nl; 3Foundation for Orthopaedic Research, Care & Education, Amphia Hospital, 4818 CK Breda, The Netherlands; ivanoost@amphia.nl

**Keywords:** fat pad sign (FPS), children, elbow, survey, receiver operating characteristic (ROC), intraclass correlation coefficients (ICC), anterior fat pad sign (AFPS), posterior fat pad sign (PFPS)

## Abstract

Children often present at the emergency department with a suspected elbow fracture. Sometimes, the only radiological finding is a ‘fat pad sign’ (FPS) as a result of hydrops or haemarthros. This sign could either be the result of a fracture, or be due to an intra-articular haematoma without a concomitant fracture. There are no uniform treatment guidelines for this common population. The aims of this study were (1) to obtain insight into FPS definition, diagnosis, and treatment amongst international colleagues, and (2) to identify a uniform definition based on radiographic measurements with optimal cut-off points via a receiver operating characteristic (ROC) curve. An online international survey was set up to assess the diagnostic and treatment strategies, criteria, and definitions of the FPS, the probability of an occult fracture, and the presence of an anterior and/or posterior FPS on 20 radiographs. Additionally, the research team performed radiographic measurements to identify cut-off values for a positive FPS, as well as test–retest reliability and inter-rater reliability via intraclass correlation coefficients (ICC). A total of 133 (paediatric) orthopaedic surgeons completed the survey. Definitions, further diagnostics, and treatments varied considerably amongst respondents. Angle measurements of the fat pad as related to the humeral axis line showed the highest reliability (test–retest ICC, 0.95 (95% CI 0.88–0.98); inter-rater ICC, 0.95 (95% CI 0.91–0.98)). A cut-off angle of 16° was defined a positive anterior FPS (sensitivity, 1.00; specificity, 0.87; accuracy, 99%), based on the respondents’ assessment of the radiographs in combination with the research team’s measurements. Any visible posterior fat pad was defined as a positive posterior FPS. This study provides insight into the current diagnosis and treatment of children with a radiological fat pad sign of the elbow. A clear, objective definition of a positive anterior FPS was identified as a ≥16° angle with respect to the anterior humeral line.

## 1. Introduction

Children often present at the emergency department with suspicion of a fracture in the elbow. The cumulative incidence of bone fracture during childhood is 40% for boys and 28% for girls between 6–16 years [1]. Approximately 28% of all paediatric fractures are elbow fractures [2]. Paediatric elbow fractures occur in approximately 25% at the distal humerus, but overall, the numbers of various types of elbow fractures vary [3,4] A prolonged time interval between injury and treatment could result in negative treatment outcomes. Unrecognised or misdiagnosed fractures presenting more than 3 weeks after the injury could result in malunion, delayed union, or non-union. [5]

The standard method of imaging are radiographs in anteroposterior and lateral directions to detect a potential fracture. Sometimes, the only finding is a fat pad sign (FPS) as a result of hydrops or haemarthros without a visible fracture. The FPS is an intra-articular, but extrasynovial, joint effusion in which the fat pad becomes elevated. Physiologically, a small anterior fat pad is usually visible. In contrast, the posterior fat pad is located in the olecranon fossa and only becomes visible in case of joint effusion. The trauma to the elbow causes intra-articular bleeding, resulting in the fat pad moving away from the joint and thus becoming visible as a triangular shape on the lateral radiographic view [6]. In such a case, the anterior and/or posterior FPS (AFPS, PFPS) could be a sign of an occult fracture [7,8].

Currently, there is no clear definition for the FPS, which may result in variations in diagnosis and treatment [9]. Usually, these children are treated with plaster and are scheduled for an orthopaedic and possibly radiographic follow-up [10]. In a number of cases, this may lead to overtreatment. Unnecessary immobilisation has disadvantages, including disability, healthcare costs, muscle atrophy, and demineralisation of the bone [10]. On the other hand, there are also fractures that require low-threshold surgical treatment, e.g., displaced lateral condyle fractures [11]. Other recent findings in the literature are the frequently missed injuries around the elbow, called the TRASH (the radiographic appearance seemed harmless) lesions [12]. Those lesions should warrant additional imaging methods, such as ultrasound, MRI, or CT, for correct diagnosis. Misdiagnosis is accountable for a large part of the complications [10]. The incidence and types of occult fractures in children with a positive FPS is difficult to assess, as the variety of definitions of FPS currently in use is high [9]. A clearer definition is mandatory to formulate a proper treatment rationale.

The aim of this study was first to assess—through an international online survey—the definition, work-up, and treatment of children with a positive FPS without visible fracture that are currently in use. Second, this study aimed to develop a clear, objective definition of an FPS that is based on radiographic measurements and international agreement on the presence of an FPS.

## 2. Materials and Methods

An online international survey was set up to assess the criteria and definitions of the AFPS and PFPS, the probability of occult fractures, diagnostic and treatment strategies, and the presence or absence of the FPS on 20 radiographs. The survey was distributed amongst members of the European Paediatric Orthopaedic Society (EPOS) and the European Society for Surgery of the Shoulder and the Elbow (SECEC), as well as national orthopaedic trauma, paediatric orthopaedic, and upper limb societies. Open questions, multiple-choice questions, and radiographs were built into the online platform of Survey Monkey (Appendix A). The study was approved by the medical research ethic committee of Utrecht (MEC-U) on 2 April 2021 under registration number W21.087.

In addition, the respondents indicated whether the AFPS and/or PFPS was present (yes/no) on 20 radiographs. The radiographs were selected from a local database of elbow radiographs of children (10 females, 7 right side, age 7–16) between 2018 and 2021, and ranged from no visible FPS to an extreme FPS without other visible pathology. Only answers of the responders who fully completed the online survey were analysed.

Furthermore, radiographic measurements were performed on the same radiographs by the research team to identify cut-off values for a positive FPS. Three paediatric orthopaedic trauma and elbow surgeons and two trained researchers (blinded to the respondents’ answers) independently measured two possible objective measures of an FPS on each of the 20 radiographs, namely, (1) the angle between the fat pad and humerus, and (2) the size of the fat pad relative to the humerus diameter (Figure 1). The averages of the five researchers’ measurements were used as the final measurement of each radiograph. The radiographs were measured again eight weeks later by the first author in a different order; the author was blinded to the first series of measurements to assess test–retest reliability.

Descriptive statistics were described for the outcomes of the survey. The test–retest and interobserver reliability of the radiographic measurements were analysed with use of intraclass correlation coefficients (ICC) [13]. The measurements with the highest ICCs (i.e., angle or distance measurements) were plotted against the percentages of positive answers to the presence or absence of an FPS by the respondents for each radiograph. This was performed in order to determine the measurement values at which the majority of professionals agreed about the presence of the AFPS and PFPS.

Further analysis was carried out with use of receiver operating characteristic (ROC) curves to determine the optimal cut-off value of a positive AFPS with corresponding sensitivity and specificity. For the purpose of the definition development, three agreement levels on the presence of an FPS were analysed with ROC curves (i.e., 50%, 60%, and 70% of the respondents). The agreement level with the highest area under the curve (AUC) was chosen to determine the optimal cut-off value. All analyses were performed using the software package SPSS (IBM SPSS, version 28, Armonk, NY, USA).

## 3. Results

There were a total of 198 respondents who started the survey, of whom 133 completed the survey entirely. The distribution of experience of the responding physicians was fairly widespread (Table 1). The most common subspecialty was paediatric orthopaedic surgery (*n* = 49).

### 3.1. Definition

The definitions of the FPS that the respondents listed varied considerably. The most commonly used terms included (sail) sign, dark shape, black shadow, elevated capsule, and darkened shade.

### 3.2. Probability of Occult Fracture

The survey showed that in the case of a positive FPS, it was expected that there is an occult fracture in an average of 73.3% (SD 18.2%) of cases. The estimated fracture rate was not significantly dependent on the respondents’ clinical experience.

### 3.3. Diagnosis and Treatment

A supracondylar humerus fracture was considered the most common fracture in case of a positive fat pad sign without visible fracture, followed by radial head and neck fractures (Table 2). Most participants used repeated radiography after 1 week in the further diagnostic work-up. Some participants replied ‘other’ (*n* = 15), mostly described by ‘depending on clinical examination’. Regarding standard treatment, the majority (*n* = 70) used plaster or a cast, but again, a wide variety was observed.

### 3.4. Radiographic Evaluation

The reliability of the angular and distance measurements of the AFPS and PFPS are shown in Table 3.

As the AFPS angle measurements (α) and the PFPS angle measurement (π) were most reliable, these were used to determine cut-off values of the proposed definitions.

An overview of radiographic angle measurements and corresponding percentages of positive answers for the presence of an AFPS is shown in Figure 2.

The analysed agreement levels of 50%, 60%, and 70% corresponded with an AUC of 0.84 (95% CI 0.64–1.00), 0.99 (95% CI 0.96–1.00), and 0.96 (95% CI 0.96–1.00), respectively. The 60% level of agreement was thus chosen to define the cut-off values of objective FPS definitions, because of the highest AUC. For the AFPS, this resulted in a cut-off value of 16° for a positive AFPS (see Figure 2).

Out of the 20 radiographs, there were 5 measurable posterior fat pads (angle π, 15.6–23.9°). The percentages of positive respondents for the visible posterior fat pads was high (72.9–100%), while those of the invisible posterior fat pads were low (0.8–34.6%). Therefore, if a posterior fat pad was visible, it was defined as positive.

The accompanying ROC curve for the objective AFPS definition of 16° had an AUC of 0.99 (95% CI 0.96–1.00) (Figure 3). Thus, the model will be able to classify 99% correctly in a given randomly selected positive AFPS and a randomly selected negative AFPS. The overall accuracy of determining the presence of a AFPS with a cut-off value of 16° was 99% (sensitivity 1.00, specificity 0.87). The ROC curve for the PFPS had an AUC of 1.00 (95% CI 1.00–1.00) (sensitivity 1.00, specificity 1.00).

## 4. Discussion

Norell first described the radiological fat pad sign of the elbow [14] in 1954. As stated in previous research, neglected paediatric elbow fractures could result in serious long-term consequences [5,11]. Today, the FPS after a traumatic event in children is still a subject within orthopaedics that lacks clarity in terms of definition, additional imaging, and guidelines for treatment. The first objective of this study was to provide more insight into the variety of definitions, work-ups, and treatments of children with a positive FPS in a large group of orthopaedic surgeons. The responses to the survey amongst the orthopaedic community varied widely. The results of the present study form an important basis in the process of developing more uniform diagnostic and treatment guidelines for children with a positive FPS without visible fracture.

Another notable result of the survey was the high expected probability of an occult fracture in the case of a positive FPS. The orthopaedic surgeons indicated, on average, that in 73.3% (SD 18.2%) of the cases with a positive FPS, they assumed the presence of an occult fracture. However, these figures are somewhat lower in the published literature. Kappelhof et al., in a meta-analysis, showed that a fracture is actually present in only 45% of children with a positive FPS and no visible fracture, based on further imaging [9].

An objective definition of a positive fat pad sign is not available in the literature [9]. Likewise, the variety in answers on definitions in the survey also highlight the need for a uniform and objective definition of an FPS. Therefore, the second aim of the study was to develop an objective definition of a positive FPS. It was found that a cut-off angle of 16° relative to the anterior humerus line indicates a positive anterior fat pad, with high reliability (ICC, 95%) and accuracy (AUC, 99%). A posterior fat pad sign was defined as any visible posterior fat pad on the lateral elbow radiograph. These findings are a first but important step towards more uniform diagnostic and treatment protocols.

A potential limitation of this study is that the development of the objective FPS definition is based on the judgements of experts with 60% agreement. At the start of the study, a range of agreements was set up in order to determine the optimal cut-off value. As described previously, determining consensus via expert panels is still a challenging element of diagnostic research [15]. By determining a cut-off value in this way, an attempt was made to do so as accurately as possible. Because of the high number of participants (n = 133), we believe these judgements reflect the general assessment of the FPS on paediatric elbow radiographs and, therefore, are the best alternative to a reference standard diagnostic method. Furthermore, the 20 radiographs, with a full range from no FPS to an extreme FPS, are considered a representative sample of the actual population. The sample could, therefore, be used reliably to assess the degree of maximum uncertainty in order to establish the optimal cut-off value for the definition of a positive FPS.

The current study provides opportunities for future research to improve the care for the investigated population. One question is whether the projection of the radiological images will affect the appearance of the FPS. Furthermore, the severity of the FPS can be studied in relation to occult fractures in children. Finally, enhanced imaging may assist physicians in optimizing patient-specific diagnoses and treatments.

## 5. Conclusions

Definitions, diagnoses, and treatments of children with a positive FPS vary considerably in the orthopaedic community. This study provides a clear and objective AFPS definition of a 16-degree angle relative to the humerus line.

## Figures and Tables

**Figure 1 children-09-00950-f001:**
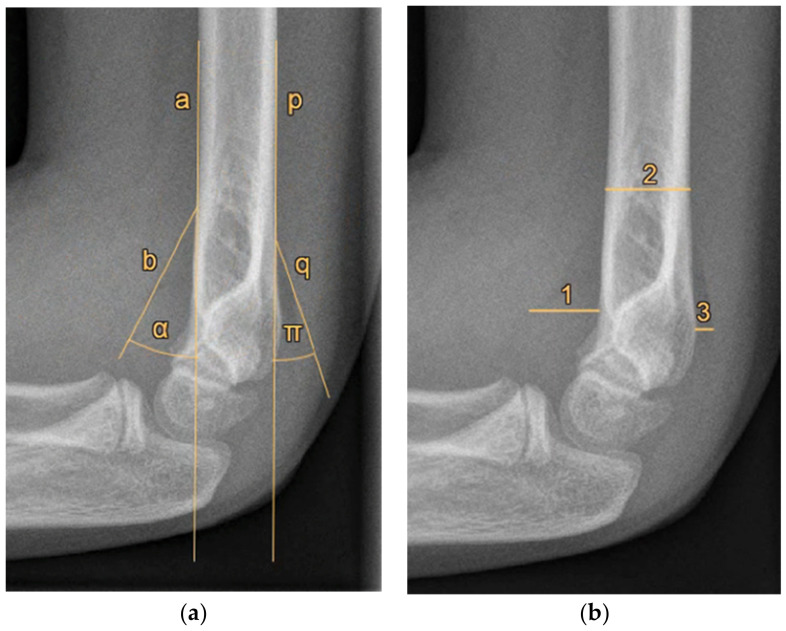
Radiographic measurements of the anterior and posterior fat pad signs. (**a**) Angle measurements. a: line along the anterior side of the humerus shaft; b: line indicating the anterior fat pad; α: angle measurement of the anterior fat pad; p: line along the posterior side of the humerus shaft; q: line indicating the posterior fat pad; π: angle measurement of the posterior fat pad. (**b**) Distance measurements—1: maximum perpendicular distance of the anterior fat pad to the humerus; 2: humerus diameter measured at the proximal level of the anterior fat pad; 3: maximum perpendicular distance of the posterior fat pad to the humerus. The anterior fat pad distance was indicated by 1 divided by 2; the posterior fat pad distance was indicated by 3 divided by 2.

**Figure 2 children-09-00950-f002:**
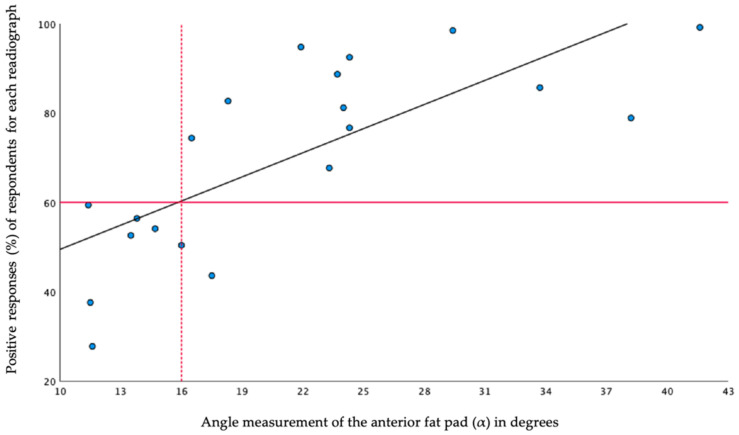
Scatter plot showing the mean angle measurements plotted against the percentages of positive respondents for the presence of an AFPS, assessed on the 20 radiographs. The horizontal red line indicates the 60% level of agreement of the respondents. The vertical dotted red line shows the mean angle at which at least 60% of the respondents indicated the presence of the AFPS.

**Figure 3 children-09-00950-f003:**
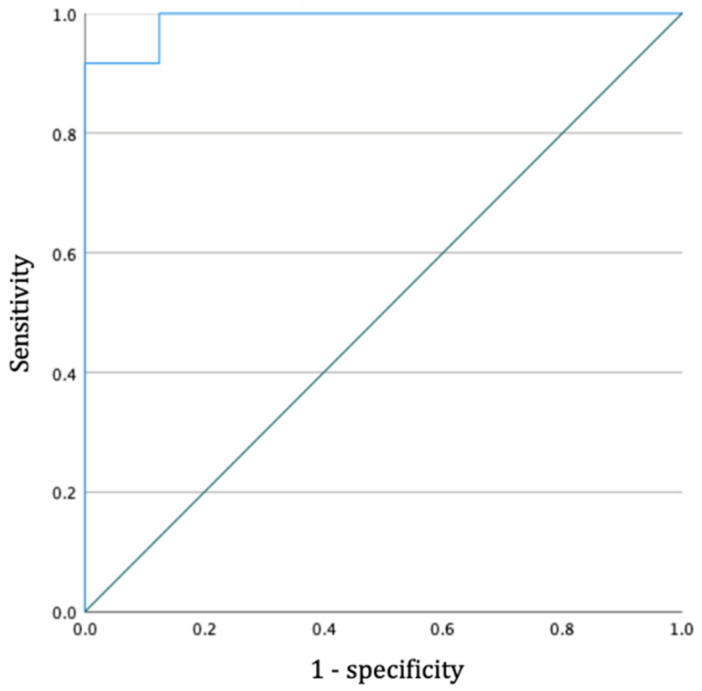
Receiver operating characteristic (ROC) curve for the 16-degree cut-off value of the anterior fat pad sign (α).

**Table 1 children-09-00950-t001:** Respondent characteristics (*n* = 133).

Survey Questions	Responses, *n* (%)
How many paediatric elbow injuries do you treat annually?	
<10	36 (27.1)
10–20	41 (30.8)
20–50	35 (26.3)
>50	21 (15.8)
How many years have you been in practice as an (orthopaedic or trauma) surgeon?	
0	14 (10.5)
1–5	31 (23.3)
6–10	18 (13.6)
>10	70 (52.6)
What is your expertise?	
Paediatric orthopaedic surgeon	49 (36.7)
Orthopaedic upper limb surgeon	28 (21.1)
Orthopaedic trauma surgeon	19 (14.3)
Orthopaedic resident	18 (13.5)
General orthopaedic surgeon	13 (9.8)
General trauma surgeon	1 (0.8)
Other	5 (3.8)

**Table 2 children-09-00950-t002:** Diagnosis and treatment (*n* = 133).

Survey Questions	Responses, *n* (%)
What is the most probable fracture in case of a positive fat pad sign without visible fracture?	
Supracondylar	87 (65.4)
Radial head	18 (13.5)
Radial neck	11 (8.3)
Lateral condyle fracture	6 (4.5)
Medial epicondyle	5 (3.8)
Other	4 (3.0)
Olecranon	2 (1.5)
What is your usual further diagnostic work-up?	
Repeat radiographs after 1 week	42 (31.6)
No further imaging	35 (26.3)
Repeat radiographs on indication	30 (22.6)
Other	15 (11.3)
CT	7 (5.2)
MRI	4 (3.0)
What is your standard treatment?	
Plaster/casting	70 (52.6)
Other	25 (18.8)
No standard treatment	11 (8.3)
Pressure bandage	11 (8.3)
Functional treatment (i.e., no immobilisation)	8 (6.0)
Sling	8 (6.0)

**Table 3 children-09-00950-t003:** Intraclass correlation coefficients (ICCs) for radiographic measurements of the anterior (AFPS) and posterior fat pad sign (PFPS).

Intraclass Correlation Coefficient		95% Confidence Interval
Anterior angle measurement (α) *		
Test–retest	0.95	0.88–0.98
Interobserver	0.95	0.91–0.98
Posterior angle measurement (π) *		
Test–retest	0.91	0.41–0.99
Interobserver	0.95	0.91–0.98
Perpendicular ratios AFPS (1/2) *		
Test–retest	0.76	0.42–0.91
Interobserver	0.74	0.38–0.89
Perpendicular ratios PFPS (3/2) *		
Test–retest	0.91	0.47–0.99
Interobserver	0.89	0.71–0.93

* See Figure 1 for corresponding measurements.

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
