# Peer review of "Diagnosis and Treatment of Children with a Radiological Fat Pad Sign without Visible Elbow Fracture Vary Widely: An International Online Survey and Development of an Objective Definition"

_children, 2022, doi:10.3390/children9070950_

Round 1

Reviewer 1 Report

I thank the Authors and the Editor for giving me the opportunity to review this manuscript. The authors carried out an international online survey to provide an objective definition of the radiological fat pad sign in children without visible elbow fracture. The article is overall interesting and well written.

Title and Abstract: OK.

Introduction: OK. I would just add a reference concerning neglected LHC fractures at line 60. (Trisolino G et al. Children (Basel). 2021 Jan 18;8(1):56. doi: 10.3390/children8010056).

Methods:

1)      I Would just specify how did you decide the number of radiographs to be assessed (N=20). Did you perform any power analysis?

2)      Moreover, it is not clear to me what the authors consider as a “true positive” or true negative FPS. Do you consider that a FPS is truly positive if 60% of the cohort of observers agree about it?

3)      Why did you decide to set the agreement level at 50%-60%-70% and not at 80%-100%?

4)      Did you confirm the presence of bone fracture by CT or X-Ray repeated after 7-10 days? Have you any information about it?

Results: OK.

Discussion: overall it is ok, but I would update it after revising the methods section.

Author Response

Reviewer 1

18 June 2022

RE: Submission to Children (ISSN 2227-9067) - 1772478

Dear Sir/Madam,

We highly appreciate your positive statements with respect to our manuscript 1772478. The text of the original manuscript has been modified in line with the valuable comments and recommendations of the reviewers, which were very helpful in improving the manuscript. An overview of the reviewers’ comments and the subsequent modifications to the manuscript (in bold italics) are listed below. A revised version with track-changes will be attached.

We hope these adjustments now make the manuscript acceptable for publication in your journal. We look forward to hearing from you in due time regarding our submission and to respond to any further questions and comments you may have.

Yours sincerely,

Max Poppelaars

Point-by-point response

We would like to thank both the reviewers for their valuable comments. Below in bold italics we reply point-by-point to the issues raised. When we refer to lines in the manuscript we refer to the highlighted version.

--------------------------

Reviewer 1 report:

Title and Abstract: OK.

No adjustments were made based on the comments of reviewer 1 on the title and abstract.

--------------------------

Introduction: OK. I would just add a reference concerning neglected LHC fractures at line 60. (Trisolino G et al. Children (Basel). 2021 Jan 18;8(1):56. doi: 10.3390/children8010056).

The reference regarding the neglected LHC fractures is very interesting and supports the foundation of this paper. I have included some information regarding neglected fractures.

--------------------------

Methods:

  • I Would just specify how did you decide the number of radiographs to be assessed (N=20). Did you perform any power analysis?

No power analysis was done regarding the number of radiographs to be assessed. Rather this amount was chosen in view of the time respondents had to put into the survey in combination with a representative dispersion of fat pad signs.

  • Moreover, it is not clear to me what the authors consider as a “true positive” or true negative FPS. Do you consider that a FPS is truly positive if 60% of the cohort of observers agree about it?

Based on the different levels of agreement, 50%, 60% and 70% respectively, the highest AUC was examined. This was the case for the 60% agreement level (AUC=99%). So with an agreement of 60%, this results in the best ratio of sensitivity to specificity for appointing a true PFPS or true FPS. Based on the response of our physicians, 60% agreement was the best level (see lines 213-217).

  • Why did you decide to set the agreement level at 50%-60%-70% and not at 80%-100%?

Given the diagnostic nature of the study, 80% - 100% agreement would result in overestimation of the fat pad definition. Therefore, as described in the methodology section, we opted for a majority of respondents. For this reason, we thought the 50, 60 and 70% levels of agreement were best. Furthermore, the AUC of the 80, 90 and 100% were lower than the AUC of the 60% level of agreement (0.99). For 80, 90 and 100& level the AUC’s were respectively 0.87, 0.84 and 0.

4)      Did you confirm the presence of bone fracture by CT or X-Ray repeated after 7-10 days? Have you any information about it?

The approach of this study was exploratory. We wanted to look at how the diagnosis and treatment of an FPS is managed, regardless of the presence of the fracture. This is definitely something we want to explore in the future.

--------------------------

Results: OK.

No adjustments were made

--------------------------

Discussion: overall it is ok, but I would update it after revising the methods section.

Adjusted (see lines 237-239).

Furthermore, some minor detail have been changed plus reference numbers.

Reviewer 2 Report

Dear Author, 

Thank you for the opportunity to review this article.

It is a manuscript that evaluates whether a radiological sign of musculoskeletal lesion without a direct sign of trauma indicates the need for treatment based on the expert opinion of 198 specialists in the field of Orthopaedics.

  1. In the introduction, You should mention TRASH elbow and You should elaborate on the severity of some occult lesions that may appear.  These are a group of osteochondral injuries having a high propensity for surgical intervention and usually have poor outcomes if not treated adequately. 
  2. An interesting fact would be to add the laterality of the patients’ hand to the subject, mentioning whether the FPS involved the dominant hand or not. Here is an article that debates hand prefference and the occurence of trauma: „The Relationship between the Dominant Hand and the Occurrence of the Supracondylar Humerus Fracture in Pediatric Orthopedics”, published in Children, DOI 10.3390/children8010051.
  3. Along with the questionnaire, did the respondents have access to patients’ clinical data? I.E. vascular injury, blunt trauma? Here is an article about neurovascular injuries that may occur in fractures about the elbow that may be useful to include in the study: „Neurovascular Abnormalities in Gartland III Supracondylar Fractures in Children”, published in Chirurgia (2013) 108: 241-244.

The article is interesting and useful in the field of Paediatric Orthopaedics.

Thank you.

Author Response

Reviewer 2

18 June 2022

RE: Your submission to Children (ISSN 2227-9067) - 1772478

Dear Sir/Madam,

We highly appreciate your positive statements with respect to our manuscript 1772478. The text of the original manuscript has been modified in line with the valuable comments and recommendations of the reviewers, which were very helpful in improving the manuscript. An overview of the reviewers’ comments and the subsequent modifications to the manuscript (in bold italics) are listed below. A revised version with track-changes will be attached.

We hope these adjustments now make the manuscript acceptable for publication in your journal. We look forward to hearing from you in due time regarding our submission and to respond to any further questions and comments you may have.

Yours sincerely,

Max Poppelaars

Point-by-point response

We would like to thank both the reviewers for their valuable comments. Below in bold italics we reply point-by-point to the issues raised. When we refer to lines in the manuscript we refer to the highlighted version.

--------------------------

Reviewer 1 report:

In the introduction, You should mention TRASH elbow and You should elaborate on the severity of some occult lesions that may appear.  These are a group of osteochondral injuries having a high propensity for surgical intervention and usually have poor outcomes if not treated adequately. 

A recent article regarding the TRASH elbow common fractures has been added to the introduction (lines 63-66).

--------------------------

An interesting fact would be to add the laterality of the patients’ hand to the subject, mentioning whether the FPS involved the dominant hand or not. Here is an article that debates hand prefference and the occurence of trauma: „The Relationship between the Dominant Hand and the Occurrence of the Supracondylar Humerus Fracture in Pediatric Orthopedics”, published in Children, DOI 10.3390/children8010051.

This information is not available in our patient history.

--------------------------

Along with the questionnaire, did the respondents have access to patients’ clinical data? I.E. vascular injury, blunt trauma? Here is an article about neurovascular injuries that may occur in fractures about the elbow that may be useful to include in the study: „Neurovascular Abnormalities in Gartland III Supracondylar Fractures in Children”, published in Chirurgia (2013) 108: 241-244.

The respondents didn’t have any information about the patients’ clinical data. This was done in order to get the opinion of the physicians regarding only the FPS.

Round 2

Reviewer 2 Report

Authors have addressed my comments.